# Low serum 25-hydroxyvitamin D status in the pathogenesis of stress fractures in military personnel: An evidenced link to support injury risk management

**Richard A. Armstrong**[1]*, **Trish Davey**[2], **Adrian J. Allsopp**[2], **Susan A. Lanham-New**[3], **Uche Oduoza**[1], **Jacqueline A. Cooper**[1], **Hugh E. Montgomery**[1‡], **Joanne L. Fallowfield**[2‡]

1 University College London Centre for Human Health and Performance and Institute for Sport, Exercise and Health, London, United Kingdom, 2 Institute of Naval Medicine, Alverstoke, Hampshire, United Kingdom, 3 Nutritional Sciences Department, Faculty of Health and Medical Sciences, University of Surrey, Guildford, Surrey, United Kingdom

‡ These authors are joint senior authors on this work.
* rarmstrong1@nhs.net

**Data Availability Statement:** Data cannot be shared publicly because they refer to patients who are members of Her Majesty's Armed Forces. Data

## Abstract

Stress fractures are common amongst healthy military recruits and athletes. Reduced vitamin D availability, measured by serum 25-hydroxyvitamin D (25OHD) status, has been associated with stress fracture risk during the 32-week Royal Marines (RM) training programme. A gene-environment interaction study was undertaken to explore this relationship to inform specific injury risk mitigation strategies. Fifty-one males who developed a stress fracture during RM training ($n = 9$ in weeks 1–15; $n = 42$ in weeks 16–32) and 141 uninjured controls were genotyped for the vitamin D receptor (VDR) *FokI* polymorphism. Serum 25OHD was measured at the start, middle and end (weeks 1, 15 and 32) of training. Serum 25OHD concentration increased in controls between weeks 1–15 (61.8±29.1 to 72.6±28.8 nmol/L, $p = 0.01$). Recruits who fractured did not show this rise and had lower week-15 25OHD concentration ($p = 0.01$). Higher week-15 25OHD concentration was associated with reduced stress fracture risk (adjusted OR 0.55[0.32–0.96] per 1SD increase, $p = 0.04$): the greater the increase in 25OHD, the greater the protective effect ($p = 0.01$). The *f*-allele was over-represented in fracture cases compared with controls ($p<0.05$). Baseline 25OHD status interacted with VDR genotype: a higher level was associated with reduced fracture risk in *f*-allele carriers (adjusted OR 0.39[0.17–0.91], $p = 0.01$). Improved 25OHD status between weeks 1–15 had a greater protective effect in *FF* genotype individuals (adjusted OR 0.31[0.12–0.81] vs. 1.78[0.90–3.49], $p<0.01$). Stress fracture risk in RM recruits is impacted by the interaction of VDR genotype with vitamin D status. This further supports the role of low serum vitamin D concentrations in causing stress fractures, and hence prophylactic vitamin D supplementation as an injury risk mitigation strategy.

sharing would breach Crown Copyright and the specific consent of these patient volunteers under the Ministry of Defence's Research Ethics Committee. Data are available via the Ministry of Defence's Research Ethics Committee (contact via the Secretariat) for researchers who meet the criteria for access to confidential Crown Copyright data. Further information and contact details can be found at https://www.gov.uk/government/groups/ministry-of-defence-research-ethics-committees.

**Funding:** Core study funding came from the United Kingdom Ministry of Defence; funds from the University College London Centre for Human Health and Performance (CHHP) supported genetic analysis. HM and the Institute for Sport, Exercise and Health (where the CHHP is housed) received funding from the National Institute for Health Research (NIHR) University College London Hospitals Biomedical Research Centre, London, UK. The funders had no role in study design, data collection and analysis, decision to publish, or preparation of the manuscript.

**Competing interests:** The authors have declared that no competing interests exist.

## Introduction

The strength and structural integrity of bone are influenced by processes that respond to changes in mechanical load (e.g. during exercise). However, repeated submaximal loading may be associated with insufficient time for bone deposition to match its removal. Bone may become weakened, and hairline stress fractures may result[1]. These injuries are common in otherwise healthy athletes and military recruits, with the reported incidence in military recruits averaging 3% in males and 9% in females[2]. The social and fiscal costs of these injuries are high, accounting for significant time off duties in military personnel[3, 4], and up to 50% of affected men and 60% of affected women fail to complete basic training[5, 6].

The molecular pathogenesis and inter-individual variability in stress fracture risk are poorly understood. However, bone strength is highly heritable[7], consistent with genetic variation influencing stress fracture risk[8–11]. Beyond smoking, lack of physical fitness, and malnourishment, no other modifiable risk factors have been consistently identified. Increasing evidence supports that low circulating vitamin D concentrations might represent one target for prophylactic intervention. The active form of vitamin D (1,25 dihydroxycholecalciferol (1,25 $(OH)_2D_3$) is a ligand for the vitamin D receptor (VDR), a transcription factor that binds to vitamin D response elements (VDREs) to regulate the expression of hundreds of genes. Thus, vitamin D plays a key role in regulating diverse biological processes and phenotypes including skeletal health[12]. Binding to the VDR on osteoblasts and osteoclasts modulates bone mineralisation and resorption[13–15]. Binding to other cells, such as hypertrophic chondrocytes, modulates their proliferation, function and survival[13]. Vitamin D also alters the lipid composition of the bone matrix[14] and the anabolic response to mechanical loading[16]. Thus, circulating vitamin D concentrations are positively associated with bone strength, cortical volume and mineral density[17–19].

Reduced vitamin D status might therefore be linked to increased stress fracture risk. Evidence supports this supposition: stress fracture risk in athletes appears inversely related to serum 25OHD status up to a concentration of 50 ng/ml (125 nmol/L)[20]. Similarly, female US Navy recruits with serum 25OHD concentrations of less than 20 ng/mL (50 nmol $L^{-1}$) had double the risk for tibia and fibula fracture compared with recruits whose circulating concentrations were at least 40 ng/mL (100 nmol/L)[21]. However, this relationship has not been demonstrated consistently, and a systematic review and meta-analysis of stress fractures in military recruits could only conclude that 'some association' appeared to exist between low vitamin D status and stress fracture risk[22]. To clarify the veracity of this association, we have previously performed a prospective study of Royal Marines (RM) recruits undertaking the 32-week RM training programme. Recruits with a baseline serum 25OHD status below 20 ng/mL (50 nmol/L) had a higher incidence of stress fracture than matched controls[23].

However, *confirmation of association* is not the same as *proof of causation*. It is possible, for instance, that a poor diet may be associated with a reduction in both vitamin D and calcium intakes, and with altered body mass index. Those taking regular exercise prior to training might also consume a diet higher in vitamin D. Seasonal variation might influence both cutaneous sunlight exposure (and hence vitamin D concentrations), and ground conditions, where the hard ground of the summer months would increase point loading of the lower limbs. An additional approach to exploring the causal nature of the association between vitamin D status and stress fracture risk is to use a genetic strategy. The gene encoding the VDR exhibits several polymorphisms that are known to be associated with the structure, metabolism and homeostasis of bone, particularly in relation to osteoporosis[24].

The single nucleotide polymorphism *FokI* produces structural differences in the VDR[25] with functional consequences: the *f* allele is associated with reduced transcription of VDR-

responsive genes compared with *FF* individuals[26]. The impact of this functional difference has been widely studied in relation to bone mineral density and osteoporosis[27–29], and the *f* allele or *ff* genotype is associated with lower concentrations of bone formation markers in US Army recruits[30].

If reduced vitamin D concentrations were causally associated with increased stress fracture risk, then it might be expected that VDR genotype would be associated with such risk. A small study in the Greek military provides some support for this conjecture: those carrying an *f* allele of the *FokI* polymorphism were at higher risk of stress fracture when compared with those of *FF* genotype (OR 4.1, 96% CI 1.3–12.7). However, vitamin D status was not reported in this study[31] and others have failed to confirm this association[8]. Furthermore, such candidate gene-association studies alone have weaknesses (and require confirmation), especially when they relate to small sample sizes[32]. A better approach would be to perform a gene-environment interaction study: evidence of an interaction between VDR genotype and circulating concentrations of its natural ligand impacting on stress fracture risk would strongly suggest a causal association between reduced circulating vitamin D concentrations and stress fracture risk in otherwise healthy, physically active, young adults.

To explore the putative causal role for low vitamin D status in stress fracture pathogenesis, we took two approaches: we performed the first gene-environment interaction study of its kind, in a convenience sample of RM military recruits who suffered stress fracture during the 32-week training programme[23], and undertook extended temporal analysis of the relationship between vitamin D and stress fracture.

## Methods

### Participants

The *Surgeon General's Bone Health Project* (SGBHP) identified risk factors for stress fracture during RM training, enrolling a total of 1,635 RM recruits. This programme was approved by the UK Ministry of Defence Research Ethics Committee (Ref: 271/Gen/11) and was conducted in accordance with the ethical standards of the Declaration of Helsinki. Details of this study have been reported previously[23, 33]. In brief, data on race, height, body weight, maximum oxygen uptake (VO$_{2max}$), smoking status, alcohol intake, physical activity and start month/season of training were recorded. Non-fasting 7 ml venous blood samples were drawn at weeks 1, 15 and 32 of training, and serum 25OHD concentration was measured using LC/MS (as described by Davey *et al*[23]). Diagnosis of stress fracture amongst recruits reporting to the Commando Training Centre Royal Marines (CTCRM) Medical Centre was confirmed by X-Ray or Magnetic Resonance Imaging (MRI) based on standard operating procedures. Of the 116 (7.1%) recruits who sustained one or more stress fractures in SGBHP, 51 (44%) were available for inclusion in this study (operational deployment, retirement from the military or killed on active service limiting access to the others). These previously stress fractured recruits were compared with 141 uninjured controls.

### Extended data analysis

We have previously reported that recruits who sustained one or more stress fractures had lower vitamin D status than those who were uninjured[23]. We performed an extended analysis to examine the temporal distribution of serum vitamin D concentration and stress fracture incidence across the 32-week training programme. For stress fracture cases, only vitamin D concentration for samples drawn before the time of injury were included. This is due to the necessary modifications to training intensity and duration of injured recruits, resulting in a

32-week vitamin D status profile that is not comparable to controls who have undergone an uninterrupted 32-week programme. The statistical methods are detailed below.

## Genetic analysis

An additional 5 ml EDTA venous blood sample was drawn from an antecubital vein in the 192 recruits. DNA was isolated and vitamin D receptor *FokI* genotype determined by polymerase chain reaction as described elsewhere[34].

## Statistical analysis

Statistical analyses were performed using the Statistical Package for Social Sciences (SPSS; Version 24, 2016).

For the extended analyses, demographic data were assessed for normality using the Shapiro-Wilk test and continuous variables compared between stress fractured and non-fractured recruits using independent *t* tests or Mann-Whitney U test, depending on distribution. Smoking, alcohol intake and regular weight-bearing activity index (number of sports played x hours per week) during the five years preceding enrolment were quantified and recruits categorised for statistical analysis. Comparisons were made using Chi-squared or Fisher's Exact Test depending on group size.

Serum vitamin D concentrations were compared using independent *t* test or Mann-Whitney U test, depending on distribution. The prevalence of vitamin D insufficiency, as per the established threshold associated with increased stress fracture risk of 50 nmol/L[23], was compared by one-tailed Chi-squared test or Fisher's Exact Test depending on group size.

For the genetic analysis, the VDR *FokI* polymorphism was analysed as a dominant model according to the presence or absence of the restriction site *f*: *f*\* or *FF*[31]. Demographic data were checked for normality using the Shapiro-Wilk test and continuous variables were compared between groups using independent *t* tests or Mann-Whitney U test, depending on distribution. Smoking habit, alcohol intake and regular weight-bearing exercise during the five years preceding military enrolment were quantified and recruits categorised for statistical analysis. Comparisons were made using Chi-squared or Fisher's Exact Test depending on group size. Genotype distribution and allele frequencies in those with and without stress fracture were compared using two-tailed Chi-squared test (or Fisher's Exact Test, depending on group size). The interaction between VDR genotype and vitamin D concentration was analysed by logistic regression before and after adjusting for age, height, weight, aerobic fitness, alcohol intake, smoking habit, physical activity level and seasonality.

To address possible bias due to the selection of the control group, controls were matched to cases using a propensity score based on 8 confounding variables: those listed above, plus date of sample. Controls were matched to cases using the MatchIt package in R[35]. Optimal matching was used as this gave the smallest mean difference between groups of the available methods. For those with complete data on all the covariates two controls were matched per case giving 33 cases and 66 controls.

To assess the impact of missing data, multiple imputation by chained equations (MICE) was used to impute 40 complete datasets. Predictive mean matching with five nearest neighbours was used for continuous variables and logistic regression for binary variables. We included all variables to be used in the analysis models including interaction terms as well as variables that correlated with the included confounders. Consistency between variables was preserved using passive imputation. Convergence and plausibility of estimates was confirmed by visual examination of plots. Propensity matching and conditional logistic regression were then performed as above for each dataset and results were combined using Rubin's rule[36].

**Table 1. Baseline characteristics of stress fracture cases and controls.**

|  |  | Cases (*n* 51) | Controls (*n* 141) | *p* |
|---|---|---|---|---|
| Age (years)[a] |  | 20 (6) | 20 (4) | 0.548 |
| White ethnicity (%) |  | 49 (96.1%) | 140 (99.3%) | 0.07 |
| Height (m)[b] |  | 1.78 (0.0086) | 1.78 (0.0047) | 0.949 |
| Weight (kg)[b] |  | 73.1 (1.0) | 75.1 (0.60) | 0.083 |
| $VO2_{max}$ (ml kg$^{-1}$ min$^{-1}$)[a] |  | 52.2 (3.8) | 52.8 (4.3) | 0.669 |
| Alcohol intake (%) | 0 | 10 (21.3%) | 17 (13.3%) | 0.450 |
|  | 1–10 units | 24 (51.1%) | 76 (59.4%) |  |
|  | 11–20 units | 11 (23.4%) | 25 (10.5%) |  |
|  | >20 units | 2 (4.3%) | 10 (7.1%) |  |
| Current smoker (%) |  | 10 (19.6%) | 34 (24.1%) | 0.512 |
| Weight bearing activity index (%) | 1 (0–19) | 17 (36.2%) | 39 (28.5%) | 0.506 |
|  | 2 (20–99) | 14 (29.8%) | 53 (38.7%) |  |
|  | 3 (>100) | 16 (34.0%) | 45 (32.8%) |  |

[a] Median (IQR)

[b] Mean (SE)

For all analyses, *p*-values <0.05 were considered statistically significant.

## Results

### Participant characteristics

The study cohort of 192 included 51 cases and 141 controls who began RM training between September 2009 and July 2010. Cases and controls did not differ in age, height, weight, $VO_{2max}$, alcohol intake, smoking status or weight bearing activity index (all $p > 0.05$, Table 1). The 51 included cases did not differ from the 65 cases unavailable for follow-up except that they had a marginally higher $VO_{2max}$. The 141 controls included had a different age distribution from those unavailable for follow-up, though the median and interquartile range were the same (see S1 and S2 Tables).

### Stress fracture timing

Nine of the 51 stress fractures (17.6%) occurred in the first half of training (weeks 1–15: 'early'), and the remaining 42 (82.4%) in the latter half (weeks 16–32: 'late') (Fig 1).

### Vitamin D status

In the control group, vitamin D status was available for 105 (of 141, 74.5%) at baseline, 78 (55.3%) at week 15 and 75 (53.2%) at week 32. The decrease in participants over time was due to operational constraints or recruits leaving training due to personal choice, medical reasons or subsequently deemed not-suitable for RM training. For stress fracture cases, only vitamin D status for samples drawn before the time of injury were included (see above). Baseline vitamin D status was available for 6 (of 9, 66.7%) 'early' cases and 34 (of 42, 81.0%) 'late' cases. Week 15 concentrations were available in 29 (69.0%) of the 'late' stress fracture group. Baseline vitamin D status was higher in those starting training in spring/summer compared with autumn/winter (78.1 + 24.9 vs 61.7 + 33.1 nmol/L, $p = 0.001$).

Baseline serum vitamin D concentrations did not differ across groups (50.4 ± 21.52 vs 63.74 ± 30.65 vs 68.61 ± 31.51 nmol/L, $p = 0.31$). In controls, serum vitamin D concentrations

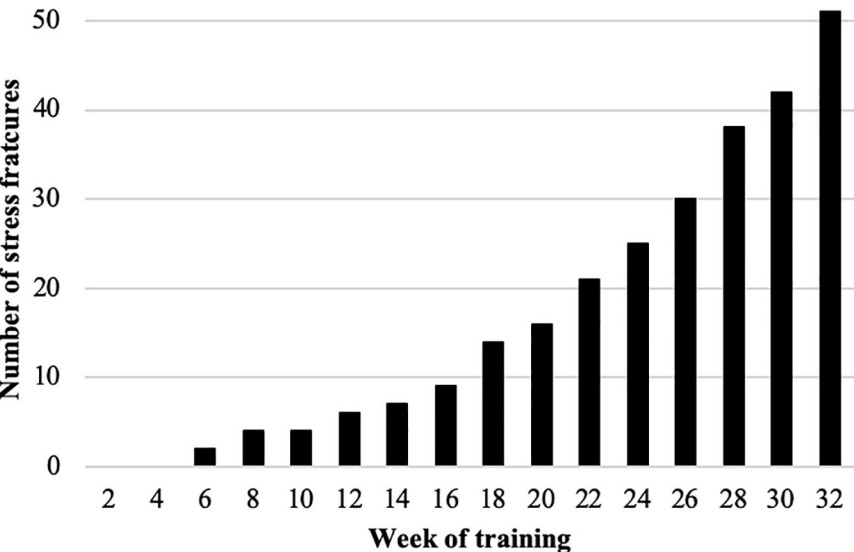

**Fig 1. Cumulative stress fracture risk.**

increased significantly by week 15 (72.6 ± 28.8 vs. 61.8 ± 29.1 nmol/L, $p = 0.014$), before falling to a value significantly below that at baseline (52.4 ± 22.7 nmol/L, $p = 0.008$). At week 15, serum vitamin D concentrations in those going on to have a stress fracture in the following weeks were similar to those at baseline (57.7 + 25.3 nmol/L, $p = 0.57$), and significantly lower at this timepoint than in controls (57.7 ± 25.3 vs. 72.6 ± 28.8 nmol/L, $p = 0.01$) (Fig 2). Higher vitamin D status at week 15 was associated with a significant reduction in 'late' stress fracture risk (adjusted OR 0.55 [0.32–0.96] per 1 SD increase in serum 25OHD, $p = 0.04$).

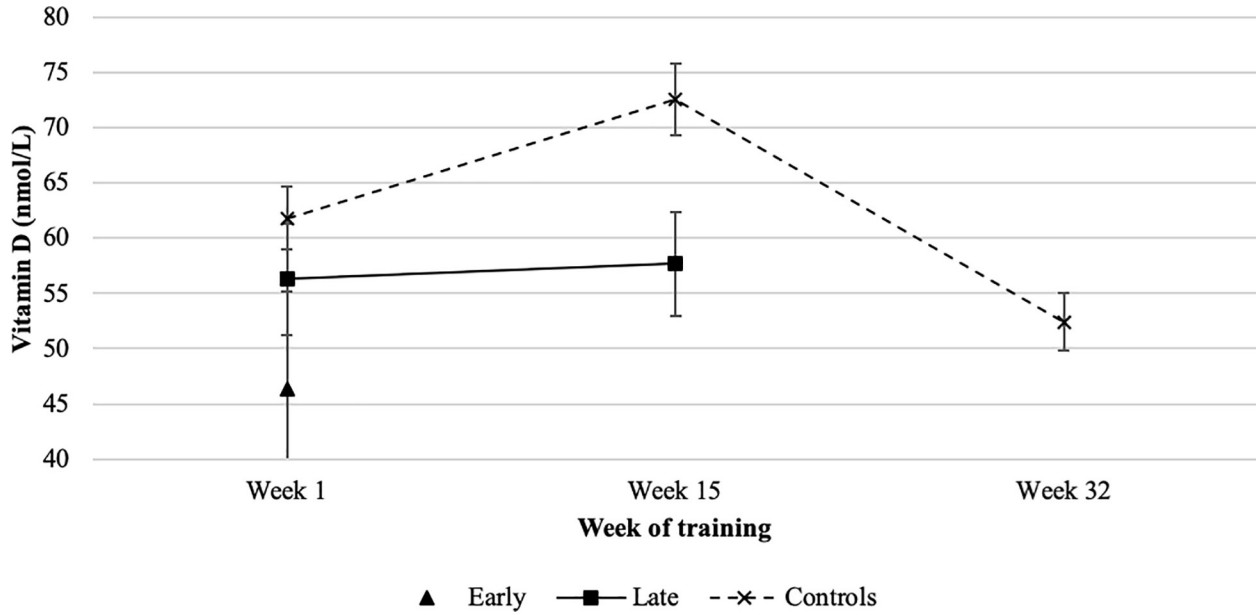

**Fig 2. Mean (+/- SEM) serum vitamin D concentration by timepoint and fracture group.**

## Vitamin D threshold

At both baseline and week 15, there was a significantly higher prevalence of vitamin D insufficiency or deficiency (defined as <50 nmol/L[37]) in the stress fracture group compared with controls (45.0% vs. 29.5%, $p = 0.04$ at baseline; 31.0% vs. 16.7%, $p = 0.04$ in week 15). The prevalence of vitamin D status below this threshold in the control group differed over the 32-week training programme, being 29.5% at baseline, 16.7% at week 15 and 46.7% at week 32 ($p = 0.0003$ overall; week 1 vs. week 15 $p = 0.04$; week 1 vs. week 32 $p = 0.02$; week 15 vs. week 32 $p = 0.00006$). In the 'late' stress fracture cases the prevalence did not differ between weeks 1 and 15 (44.1% vs. 31.0%, $p = 0.29$).

## Vitamin D receptor genotype and allele distribution

No differences in demographic data or the season in which training commenced were found across genotypes (S3 and S4 Tables). There were no differences in genotype distribution or allele frequencies when comparing all stress fractured recruits to non-fractured recruits (*FF* 15 (29.4%) vs. 52 (36.9%), *Ff* 29 (56.9%) vs. 65 (46.1%), *ff* 7 (13.7%) vs. 24 (17.0%), $p = 0.21$; allele frequencies *F* 59 (57.8%) vs. 169 (59.9%), *f* 43 (42.2%) vs. 113 (40.1%), $p = 0.36$). The same was true when comparing 'early' cases, 'late' cases and controls (*FF* 0 (0%) vs. 15 (35.7%) vs. 52 (36.9%), *Ff* 7 (77.8%) vs. 22 (52.4%) vs. 65 (46.1%), *ff* 2 (22.2%) vs. 5 (11.9%) vs. 24 (17.0%), $p = 0.07$; allele frequencies *F* 7 (38.9%) vs. 52 (61.9%) vs. 169 (59.9%), *f* 11 (61.1%) vs. 32 (38.1%) vs. 113 (40.1%), $p = 0.09$). Similarly, there were no differences in genotype or allele distribution between 'late' cases and controls ($p > 0.05$). However, there was a significant excess of *f*-containing genotypes and thus of the *f* allele in 'early' stress fracture cases compared with controls ($p = 0.04$ for genotype, $p = 0.008$ for alleles).

## Vitamin D status, receptor genotype and stress fracture risk

Logistic regression for main effects showed that overall there was no significant association of either serum vitamin D concentration or VDR genotype with stress fracture ($p = 0.20$ and 0.70, respectively).

## VDR genotype-baseline vitamin D interaction

The interaction between genotype and baseline vitamin D concentration was significant (Table 2). In *f*-allele carriers, higher baseline vitamin D status was associated with a reduction in stress fracture risk (adjusted OR 0.39 [0.17–0.91], $p = 0.01$).

**Table 2. Logistic regression for stress fracture including vitamin D receptor (VDR) x baseline vitamin D status interaction.**

| VDR genotype (n) | OR per 1 SD increase in baseline vitamin D status | P | Adjusted§ | p |
|---|---|---|---|---|
| *FF* (51) | 1.64 (0.80–3.34) | 0.049 | 2.02 (0.71–5.71) | 0.045 |
| *Ff* (71) | 0.65 (0.35–1.18) | | 0.37 (0.14–0.96) | |
| *Ff* (23) | 0.31 (0.08–1.26) | | 0.46 (0.10–2.11) | |
| *FF* (51) | 1.64 (0.80–3.34) | 0.02 | 2.03 (0.72–5.69) | 0.01 |
| *f** (94) | 0.55 (0.32–0.94) | | 0.39 (0.17–0.91) | |

§adjusted for age/height/weight/VO2max/alcohol/smoking/activity/seasonality. P values refer to the interaction term.

**Table 3. Logistic regression for late stress fracture risk–including vitamin D receptor (VDR) x vitamin D status increase interaction.**

| VDR genotype (n) | OR per 1 SD increase from baseline vitamin D status | p | Adjusted§ | p |
|---|---|---|---|---|
| FF (39) | 0.48 (0.22–1.06) | 0.04 | 0.31 (0.12–0.81) | 0.005 |
| f* (71) | 1.31 (0.76–2.26) | | 1.78 (0.90–3.49) | |

§for age/height/weight/VO2/alcohol/smoking/activity/seasonality. P values refer to the interaction term.

## Propensity matching

Thirty-three case participants had both baseline vitamin D status and complete covariate data and were propensity matched in a 1:2 ratio to 66 controls. After matching, the balance between groups was improved with the standardised mean difference in distance reduced from 0.49 to 0.19. An absolute value below 0.2 was observed for all variables after matching indicating that any imbalance between the groups was small[38]. There was no significant association of either vitamin D status or genotype alone with stress fracture risk ($p$ = 0.51 and 0.16, respectively). However, there was a significant interaction between genotype and baseline vitamin D concentration: higher serum 25OHD concentration was protective in $f$-allele carriers but not in the *FF* group (S5 Table).

## Multiple imputation

The data had missing values for vitamin D status (24%), smoking habit (8%), alcohol consumption (9%), physical activity (4%) and $VO_{2max}$ (7%). Odds ratios were consistent before and after imputation, and the interaction term remained significant ($p$ = 0.05) indicating that the missing data had not caused bias in the results (S6 Table).

## Change in vitamin D status from week 1 to week 15

A greater increase in vitamin D concentration from baseline to week 15 was associated with a reduction in risk of 'late' stress fracture after adjusting for genotype, baseline vitamin D status, demographic data and seasonality (OR 0.77 [0.61–0.97] per 10 unit increase in vitamin D status, $p$ = 0.03). For a given increase in vitamin D concentration between week 1 and week 15, a greater protective effect against 'late' stress fractures was seen in the *FF* group than in $f$-allele carriers (adjusted OR 0.31 [0.12–0.81] vs. 1.78 [0.90–3.49] per 1 SD increase, $p$ = 0.005) (Table 3).

## Discussion

Temporal changes in circulating vitamin D concentration were associated with stress fracture risk in RM recruits undertaking the 32-week RM training programme. The prevalence of vitamin D insufficiency or deficiency (defined as <50 nmol/L) was consistently greater in the stress fracture group. These data are consistent with previous reports of similar associations. However, the importance of the present study is that, in adopting a gene-environment interaction measurement approach, evidence is presented that suggests that the association of low serum vitamin D concentration with stress fracture risk is likely causal. Moreover, this study also indicates that the efficacy of a vitamin D supplementation strategy to mitigate stress fracture risk will be dependent upon both vitamin D status and a recruit's vitamin D receptor *FokI* genotype.

Baseline vitamin D status did not differ between those recruits who sustained a stress fracture and those who did not, in keeping with our previous findings[23]. However, mean vitamin D concentrations beyond this point showed a different trajectory in controls compared

with those who stress fractured in the latter half of training (weeks 16–32). At the midpoint of training (week 15) controls had a higher mean 25OHD concentration than those who went on to fracture 'late' (77.71 vs. 62.76 nmol/L, $p$ = 0.01), having increased significantly from their baseline vitamin D concentration (72.6 vs. 61.8 nmol/L, $p$ = 0.014). In contrast, those who fractured in weeks 16–32 of RM training did not show any increase from baseline concentrations at the midpoint of training (56.3 vs. 57.7 nmol/L, $p$ = 0.57). Furthermore, a greater serum 25OHD concentration in week 15 was associated with a significant reduction in 'late' stress fracture risk, whilst a greater magnitude of increase in vitamin D concentration between baseline and week 15 gave a greater protective effect (adjusted OR 0.77 [0.61–0.97] per 10 unit increase, $p$ = 0.03). These results suggest that–to mitigate injury risk–vitamin D status must not only be maintained, but must also *increase* over the early part of training to meet the increased training demands and physical loading to which bone is exposed.

## Vitamin D concentration threshold

Previous work in RM recruits has demonstrated an increased risk of stress fracture when baseline vitamin D concentration is below a threshold of 50 nmol/L[23]. This would be classified as 'insufficient'[37] and is recognised by the UK *National Osteoporosis Society* as potentially inadequate[39]. The prevalence of vitamin D status below that level in this cohort was 33.8% at baseline, falling to 20.6% in those uninjured by week 15, and then rising to a peak of 46.7% in controls at the end of the 32 week training programme. Controls also showed significantly lower circulating vitamin D concentrations at the end of training compared with baseline (52.4 vs. 61.8 nmol/L, $p$ = 0.008), with a higher prevalence of vitamin D insufficiency or deficiency (46.7% vs. 29.5%, $p < 0.001$). This is in keeping with both our previous work[23] and reports of female recruits undertaking US Army basic combat training[40], and is likely multifactorial in origin. Circulating serum vitamin D concentrations reflect the combined influence of dietary intake and cutaneous synthesis, both of which may be modified during military training.

Together, these data suggest that vitamin D supplementation has the potential to reduce some of the inherent baseline stress fracture risk. To our knowledge, there have been no interventional trials of vitamin D alone to reduce stress fracture in military recruits. A randomised placebo-controlled trial of calcium and vitamin D in female US Navy recruits resulted in a 20% lower incidence of stress fracture. However, vitamin D status was not measured and so the attribution of the effect was unclear[6]. US Army recruits randomised to calcium and vitamin D (2000 mg and 1000 IU per day, respectively) or placebo, increased vitamin D over training in both groups. However, the intervention group had higher vitamin D status at baseline, stress fracture incidence was not reported, the training period was shorter at only 9 weeks, and the individual contributions of calcium and vitamin D could not be delineated[41].

The observed effect of the temporal course of vitamin D status throughout training suggests that the relationship with stress fracture risk may not be so straightforward. Given the seemingly non-linear relationship between vitamin D status and stress fracture risk, it is likely that either there is a required threshold for action and/or that vitamin D status *per se* is not the sole contributor.

## Vitamin D receptor

The VDR genotype distribution and allele frequencies did not differ between stress fracture cases and controls. This contrasts with a study of 32 military recruits in which stress fracture cases were more likely to have the *Ff* or *ff* genotypes, with the presence of the *f*-allele associated with increased stress fracture risk[31]. Vitamin D status was not reported, and hence it may be

that the gene-environment interaction is the ultimate determinant of risk. When separated by time of injury, our data did demonstrate an excess of *f*-containing genotypes in those recruits who fractured 'early' compared with controls (100% vs. 63.1%, *p* = 0.02) and 'late' fracture cases (100% vs. 64%, *p* = 0.04); those recruits who fractured in the latter half of training did not differ from controls. This suggests that there is a distinct cohort who may be prone to early injury, and this is consistent with the above study. However, the sample size of 'early' stress fracture cases in the present study was small (n = 9 recruits).

## Vitamin D receptor gene-environment interaction

Analysed in isolation, neither VDR genotype nor vitamin D status were associated with stress fracture risk (*p* > 0.05). However, the VDR-baseline vitamin D *interaction* was significant: in *f*-containing genotypes, increased baseline vitamin D status was associated with reduced stress fracture risk (adjusted OR 0.39 [0.17–0.91], *p* = 0.01), but this effect was not seen in *FF* individuals. This association remained after propensity matching and imputation of missing data, indicating that neither the selection of the control group nor missing data caused bias in the results.

As noted above, an increase in vitamin D status between baseline and week 15 was associated with a reduction in stress fracture risk later in training. However, this effect varied depending on VDR genotype: for a given increase in vitamin D status between weeks 1 and 15, *FF* genotype individuals showed a greater reduction in later stress fracture risk than *f*-containing genotypes (adjusted OR 0.31 [0.12–0.81] vs. 1.78 [0.90–3.49], *p* = 0.005).

Thus, for *f*-containing genotypes, baseline vitamin D concentration was a significant determinant of stress fracture risk in training, with the magnitude of change from weeks 1 to 15 relatively less important. This suggests that the initial response to the training load differs across genotypes and a baseline vitamin D status that may be adequate in one individual may be insufficient for another. This is an important observation to consider when planning a vitamin D supplementation regimen to mitigate injury risk.

The absence of a protective effect with a greater increase in vitamin D status between weeks 1 and 15 in *f*-containing genotypes may represent an inability to respond fully to increasing vitamin D concentrations due to a less efficient VDR. This is supported by evidence from US Army recruits undergoing initial military training who displayed lower concentrations of bone formation markers in response to training with *f*-containing genotypes[30]. Additionally, the *f*-containing genotypes are known to be associated with reduced transcription of VDR-responsive genes[26] and it may be that relatively more vitamin D is required to undertake the bone remodelling that training requires. In contrast, *FF* individuals are able to adequately respond at a lower vitamin D concentration. A gene-environment interaction has previously been reported in prostate cancer, with the *ff* genotype associated with increased risk of prostate cancer only in those who had a plasma vitamin D concentration below the median; with a plasma vitamin D concentration above the median, *ff* was no longer associated with increased risk. This suggests that the status of vitamin D sufficient to prevent cancer in one person may be insufficient in another individual with a different genotype[42]. It may be that a similar paradigm is operating in this study, where a baseline vitamin D status sufficient to protect against stress fracture in *FF* individuals is insufficient in *f*-allele carriers.

The balance between the physiologically active form of vitamin D, $1,25(OH)_2D_3$, and its precursor 25(OH) is known to be affected by VDR genotype, with carriers of the *F* allele having higher ratios of the active to inactive form[43]. Hence the physiological effects of a given vitamin D status may vary according to VDR genotype and this may add to the explanation of the difference in stress fracture risk observed in the present study.

These results demonstrate, for the first time, that it is the *gene-environment* interaction, *specifically* between the VDR genotype and the natural ligand vitamin D, which dictates risk of stress fracture in otherwise healthy individuals, rather than either factor in isolation. This finding may help explain the presence of conflicting data relating vitamin D status with stress fracture risk in other studies. Furthermore, stress fracture risk (and the nature of the gene-environment interaction) changed as training progressed, and all three factors should now be considered in the design of such studies, and–importantly for military planners–the design of injury risk mitigation interventions.

## Limitations

The analyses reported here were only possible in 44% of recruits who sustained a stress fracture during the course of the SGBHP, and as a result our sample size is small, particularly for those who sustained an 'early' fracture. Similarly, controls were location-matched due to pragmatic limitations, although propensity matching indicated that this had not introduced bias. Whilst we have adjusted for a number of confounding factors, it is acknowledged that there may be other potential confounders for which data were not collected. As described, only one genetic polymorphism related to stress fracture risk was examined, where several novel candidate genes have since been reported[44]. The recruits included in this study had marginally higher baseline fitness that those who were not followed-up, which may be explained by the fact that only those still in service after seven years were eligible. The vast majority of RM recruits in this study were of white ethnic origin, and at the time of the study RM recruit training was only open to males. As such, these findings may not be as applicable to different ethnic groups or females. Finally, RM training is an intensive 32-week programme that differs to those undertaken by other military personnel.

## Conclusions

These data offer further insight into our earlier finding that a lower baseline vitamin D concentration was associated with an increased risk of stress fracture in RM recruits[23]. First, they highlight the effect that an individual's vitamin D status over the course of training has on stress fracture risk, beyond that imposed by the baseline 25OHD concentration alone, suggesting a possible role for a prophylactic intervention. Second, they demonstrate that stress fracture risk in response to vitamin D concentration is dependent upon individual genotype, raising the possibility that vitamin D supplementation may need to be individualised (elevated) in some. The data presented here provide further evidence in support of a therapeutic trial of vitamin D supplementation in RM recruits in training to reduce the risk of stress fracture. In view of the gene-environment interaction outlined above, any such trial should include individuals with both low and normal baseline vitamin D status.

## Supporting information

**S1 Table. Baseline characteristics of stress fracture cases available and unavailable for follow-up.** [a]Median (IQR) [b]Mean (SE).
(DOCX)

**S2 Table. Baseline characteristics of controls available and unavailable for follow-up.** [a]Median (IQR) [b]Mean (SE).
(DOCX)

**S3 Table. Baseline participant characteristics across vitamin D receptor (VDR) genotypes.**
[a]Median (IQR) [b]Mean (SE).
(DOCX)

**S4 Table. Start of training across vitamin D receptor (VDR) genotypes.**
(DOCX)

**S5 Table. Logistic regression for stress fracture risk in propensity matched cases and controls–vitamin D receptor (VDR) genotype x baseline vitamin D interaction.**
(DOCX)

**S6 Table. Logistic regression for stress fracture risk after multiple imputations–vitamin D receptor (VDR) genotype x baseline vitamin D interaction.**
(DOCX)

## Acknowledgments

The authors would like to thank colleagues at the Commando Training Centre Royal Marines, Lympstone, Devon, UK, for their cooperation and support with this study.

## Author Contributions

**Conceptualization:** Trish Davey, Adrian J. Allsopp, Susan A. Lanham-New, Hugh E. Montgomery, Joanne L. Fallowfield.

**Data curation:** Richard A. Armstrong, Trish Davey, Adrian J. Allsopp, Susan A. Lanham-New, Jacqueline A. Cooper.

**Formal analysis:** Richard A. Armstrong, Jacqueline A. Cooper.

**Funding acquisition:** Adrian J. Allsopp, Hugh E. Montgomery, Joanne L. Fallowfield.

**Investigation:** Richard A. Armstrong, Trish Davey, Adrian J. Allsopp, Susan A. Lanham-New, Hugh E. Montgomery.

**Methodology:** Richard A. Armstrong, Trish Davey, Adrian J. Allsopp, Susan A. Lanham-New, Jacqueline A. Cooper, Hugh E. Montgomery, Joanne L. Fallowfield.

**Project administration:** Trish Davey, Adrian J. Allsopp, Joanne L. Fallowfield.

**Supervision:** Adrian J. Allsopp, Susan A. Lanham-New, Hugh E. Montgomery, Joanne L. Fallowfield.

**Visualization:** Richard A. Armstrong.

**Writing – original draft:** Richard A. Armstrong, Trish Davey, Adrian J. Allsopp, Susan A. Lanham-New, Uche Oduoza, Jacqueline A. Cooper, Hugh E. Montgomery, Joanne L. Fallowfield.

**Writing – review & editing:** Richard A. Armstrong, Trish Davey, Adrian J. Allsopp, Susan A. Lanham-New, Uche Oduoza, Jacqueline A. Cooper, Hugh E. Montgomery, Joanne L. Fallowfield.

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
