## [Decision Letter · Decision Letter 0]

12 Feb 2020

Low serum 25-hydroxyvitamin D status in the pathogenesis of stress fractures in military personnel: an evidenced link to support injury risk management

PONE-D-19-29378

Dear Dr. Armstrong,

We are pleased to inform you that your manuscript has been judged scientifically suitable for publication and will be formally accepted for publication once it complies with all outstanding technical requirements.

With kind regards,

Giuseppe Sergi

Academic Editor

PLOS ONE

**Comments to the Author**

1. Is the manuscript technically sound, and do the data support the conclusions?

Reviewer #1: Yes

Reviewer #2: Yes

2. Has the statistical analysis been performed appropriately and rigorously? 

Reviewer #1: Yes

Reviewer #2: Yes

3. Have the authors made all data underlying the findings in their manuscript fully available?

Reviewer #1: Yes

Reviewer #2: Yes

4. Is the manuscript presented in an intelligible fashion and written in standard English?

Reviewer #1: Yes

Reviewer #2: No

5. Review Comments to the Author

Reviewer #1: The authors aimed to explore the association between the serum 25-hydroxyvitamin D status and vitamin D receptor polymorphism in military recruits with and without stress fractures during a 32-week training program. The question posed by the authors is well defined.

Data and its analysis seem appropriate to my understanding. The limitations are clearly addresses, conclusions are well balanced.

Reviewer #2: You completed a valuable review which gives further guideline to future practice.

6. PLOS authors have the option to publish the peer review history of their article (what does this mean?). If published, this will include your full peer review and any attached files.

Reviewer #1: Yes: Albrecht W Popp, MD, Bern University Hospital, Switzerland

Reviewer #2: Yes: Leung Ping-chung

---

## [Editor Report · Acceptance letter]

14 Feb 2020

PONE-D-19-29378 

Low serum 25-hydroxyvitamin D status in the pathogenesis of stress fractures in military personnel: an evidenced link to support injury risk management 

Dear Dr. Armstrong:

I am pleased to inform you that your manuscript has been deemed suitable for publication in PLOS ONE. Congratulations! Your manuscript is now with our production department. 

With kind regards,

on behalf of

Dr. Giuseppe Sergi 

Academic Editor

PLOS ONE